# Raman Spectroscopy in Prostate Cancer: Techniques, Applications and Advancements

**DOI:** 10.3390/cancers14061535

**Published:** 2022-03-17

**Authors:** Fortis Gaba, William J. Tipping, Mark Salji, Karen Faulds, Duncan Graham, Hing Y. Leung

**Affiliations:** 1Department of Urology, Queen Elizabeth University Hospital, NHS Greater Glasgow and Clyde, Glasgow G51 4TF, UK; fortis.gaba@ggc.scot.nhs.uk (F.G.); mark.salji@ggc.scot.nhs.uk (M.S.); 2School of Medicine, University of Glasgow, Glasgow G12 8QQ, UK; 3Department for Pure and Applied Chemistry, University of Strathclyde, Glasgow G1 1RD, UK; william.tipping@strath.ac.uk (W.J.T.); karen.faulds@strath.ac.uk (K.F.); duncan.graham@strath.ac.uk (D.G.); 4Institute of Cancer Sciences, College of Medical, Veterinary and Life Sciences, University of Glasgow, Glasgow G61 1QH, UK; 5CRUK Beatson Institute, Bearsden, Glasgow G61 1BD, UK

**Keywords:** Raman spectroscopy, prostate cancer, diagnostics, therapeutics, biomarker

## Abstract

**Simple Summary:**

Raman spectroscopic analysis is a promising and powerful optical investigative tool to interrogate tissue and/or biofluids for in vitro and in vivo applications. Fast-acquisition Raman imaging techniques support real-time, objective evaluation of molecular tissue composition in prostate cancer. Within the capability of non-destructive and non-invasive imaging, Raman spectroscopy-based techniques offer a reliable tool for cancer detection. There is a push for intraoperative use and/or multimodal imaging approaches that utilise the strength of each spectroscopic modality to improve prostate cancer detection and treatment.

**Abstract:**

Optical techniques are widely used tools in the visualisation of biological species within complex matrices, including biopsies, tissue resections and biofluids. Raman spectroscopy is an emerging analytical approach that probes the molecular signature of endogenous cellular biomolecules under biocompatible conditions with high spatial resolution. Applications of Raman spectroscopy in prostate cancer include biopsy analysis, assessment of surgical margins and monitoring of treatment efficacy. The advent of advanced Raman imaging techniques, such as stimulated Raman scattering, is creating opportunities for real-time in situ evaluation of prostate cancer. This review provides a focus on the recent preclinical and clinical achievements in implementing Raman-based techniques, highlighting remaining challenges for clinical applications. The research and clinical results achieved through in vivo and ex vivo Raman spectroscopy illustrate areas where these evolving technologies can be best translated into clinical practice.

## 1. Introduction

Prostate cancer is the second most frequent malignancy in men and the fifth leading cause of death worldwide [1]. The principal challenge for healthcare is to understand the needs of this increasing group of men, in particular, utilising new technologies for improved cancer detection. Numerous research groups have investigated the potential of Raman spectroscopy for clinical use. In this paper, we will review the theory and concepts of Raman scattering, the development of enhanced Raman imaging techniques and sample preparation and analysis (Table 1). In addition, we will review the clinical application of Raman spectroscopy relevant to the field of prostate cancer.

## 2. Principles of Raman Spectroscopy

Raman spectroscopy is a powerful analytical methodology that can characterise biological specimens at high spatial resolution in a non-invasive manner. To do so, Raman spectroscopy uses monochromatic light, typically generated from a laser source in the mid-ultraviolet (mid-UV, 200–400 nm), visible (400–700 nm) or near-infrared (NIR, 0.7–1.1 μm) regions. Upon excitation of the sample by an incident laser, two scattering phenomena can occur: Rayleigh scattering (elastic scattering) and Raman scattering (inelastic scattering) (Figure 1A). Of these, elastic scattering is dominant and produces photons of the same energy, and thus wavelength and frequency, as the incident photons. Since there is no transfer of energy from the incident photons, either to or from the molecule under investigation, Rayleigh scattering does not yield any specific information about the molecule, and it is hence detected at the excitation wavelength (Raman shift = 0 cm^−1^). Alternatively, the light may be scattered at a range of different wavelengths, and hence frequencies, in a process referred to as Raman scattering (or inelastic scattering). Raman scattering is a rare phenomenon, with only ~1 in 10^8^ incident photons undergoing inelastic scattering. The inelastic scattered photons will have either higher energy (anti-Stokes Raman scattering) or lower energy (Stokes Raman scattering) due to the interaction of incident photons with vibrational modes associated with chemical bonds within the target sample. The energy difference between the incident photons and the Raman scattered photons characterises the chemical bonds excited by that energy and is measurable as a difference in wavelength of light compared to the excitation wavelength. As such, a Raman spectrum is a plot of the photon intensity in arbitrary units (a.u. or arb. units) as a function of Raman shift, which is plotted in wavenumber, cm^−1^ (1/wavelength). A Raman spectrum, therefore, characterises the various vibrational modes associated with chemical bonds that comprise the sample. The intensity of vibrations produced in Raman spectroscopy is dependent on the Raman cross-section of the molecule being interrogated. Water has a low Raman cross-section, meaning that aqueous samples are ideal for analysis by Raman spectroscopy due to the low background from the water; however, sensitivity can be an issue and is discussed later in this review.

Compensating for the low probability associated with Raman scattering, high-efficiency laser sources, high-throughput optics and low-noise detectors have enabled Raman spectroscopy to be developed into a powerful imaging technique capable of reporting on the underlying biology of the sample. Interfacing Raman spectroscopy with an optical microscope has enabled the development of Raman imaging using high-magnification lenses to deliver subcellular spatial resolution (Figure 1B). The spatial resolution of optical Raman microscopy is diffraction-limited and therefore is dependent upon the excitation laser wavelength (λ) and the numerical aperture of the objective lens. Modern Raman microscopes can achieve sub-micron resolution, although in practice, it is not possible to reach the diffraction limit due to light scattering at the sample interface and inefficiencies in light throughput via optical components. Typically, ~500 nm resolution can be achieved using commercially available instruments [2]. Table 2 highlights some of the key advantages and disadvantages of spontaneous Raman scattering for biomedical applications.

The sensitivity of Raman scattering is dependent on the excitation wavelength, where the Raman scattering intensity is proportional to 1/λ^4^. In practice, a laser with a short wavelength will have a greater intensity of Raman scattering and hence sensitivity than a laser with a long wavelength. However, the lower photon energy associated with the increasing laser wavelength can be advantageous for biological imaging, due to reduced sample phototoxicity which can result in sample overheating and burning. In addition, background absorption and fluorescence from tissue samples are generally reduced when NIR lasers are used for imaging, which reduces the background autofluorescence from the cell/tissue in the resultant spectra and images. The use of pulsed laser sources can also promote a similar effect. For most biological applications, the laser power focused onto the sample is typically <50 milliwatts (mW) for mid-UV and visible laser sources and <200 mW for NIR laser sources.

The Raman spectra of biological samples may contain diagnostic peaks within the fingerprint region (typically 400–1800 cm^−1^) associated with proteins, lipids, DNA and carbohydrates [3]. The high-wavenumber region (2700–3500 cm^−1^) of the Raman spectrum consists largely of C-H, N-H and O-H stretching modes of proteins, lipids and DNA, among other biological species. Notably, the region between 1800 and 2600 cm^−1^ is largely free from endogenous cellular/biomolecular peaks and is therefore referred to as the cell-silent region. A representative Raman spectrum of live human PC-3 prostate cancer cells is shown in Figure 2, which shows a large number of characteristic peaks in the high-wavenumber region (>2800 cm^−1^) and the fingerprint region (<1800 cm^−1^). Labelling strategies have successfully exploited the cell-silent region for discrete imaging of labelled biomolecules using alkyne, nitrile and deuterium functional groups, which all have diagnostic Raman shifts within this region of the spectrum [4]. The detection of these functional groups within the cell-silent region of the Raman spectrum offers a high degree of signal specificity compared to the endogenous cellular Raman signals. Ratiometric Raman imaging of live cells has been described previously [5], and the ratiometric analysis of live PC-3 cells is presented in Figure 2. These images readily delineate the nuclear regions from the lipid-rich cytoplasm regions.

## 3. Enhanced Raman Techniques

A variety of enhanced Raman techniques have been developed for greater sensitivity detection, enhanced depth profiling and faster imaging acquisition. The following sections will introduce the concepts behind several techniques that have been directly applied to study prostate cancer. Table 3 highlights some of the advantages and limitations of each of these techniques.

### 3.1. Resonance Raman Scattering (RRS)

Resonance Raman scattering (RRS) occurs when the frequency of the incident laser matches, or is close to, the frequency of an electronic transition of the target sample, which results in scattering enhancements in the region of 10^3^ to 10^4^ over spontaneous Raman scattering. Given that only the selected chromophore is in resonance, enhanced Raman scattering is produced with improved sensitivity and selectivity compared to normal Raman scattering. For instance, by applying RRS, the release of cytochrome C can be monitored based on a 532 nm wavelength excitation laser in resonance with the heme group to allow detection of apoptosis in live cells in real time [6].

### 3.2. Surface-Enhanced Raman Scattering (SERS)

Surface-enhanced Raman scattering (SERS) involves the interaction of the analyte molecule or a Raman reporter with a roughened metal surface, which affords an electromagnetic enhancement of the signal in the order of 10^4^ to 10^8^ over spontaneous Raman scattering. Nanoparticles (typically <100 nm) that are formed from noble metals (e.g., gold and silver) are typically used, which act as an antenna to amplify the Raman scattering signal of the analyte that is in close proximity to the metallic surface (Figure 3A). Gold and silver nanoparticles are optimal SERS substrates due to their unique optical properties and adaptable synthesis that allows control over size, shape and morphology, enabling flexibility to tailor the nanoparticles for diagnostic applications [7]. SERS-active nanoparticles can either be used in a label-free (direct) capacity, where the biomolecule of interest is directly adsorbed onto a nanoparticle surface and the intrinsic SERS signal is obtained, or for labelled (indirect) SERS detection, where Raman reporter molecules (for example malachite green) are added to the nanoparticle surface to create SERS nanotags that can be used to indirectly detect biomolecules [7]. Typically, the Raman reporter molecule is a chromophore with an electronic transition that matches, or is close to, the energy of the excitation wavelength. This results in signal enhancement via resonance effects and is referred to as surface-enhanced resonance Raman scattering (SERRS). SE(R)RS has been used in a variety of applications, including label-free analysis for the diagnosis of prostate cancer [8], the detection of prostate-specific antigen using SERS in a microfluidic device [9] (Figure 3B) and for prostate cancer disease stratification (Figure 3C) [10]. With the enhanced signals generated through the use of SERS-active nanoparticles, there is significant interest in their development for in vivo detection of disease. By capitalising on the resonance enhancement effects, SERRS has been used to provide accurate delineation of tumour margins and enabled lymph node metastases in a preclinical prostate cancer model [11]. SERRS has potential for clinical translation given that: (i) the components of SERRS nanoparticles are typically inert materials, (ii) some nanoparticles have been previously identified as non-toxic in vivo [12] and (iii) nanoparticles have even advanced into clinical trials [13]. For in vivo delivery, SERS nanoparticles have been injected intravenously via the tail vein of tumour-bearing mice, and Raman analysis was conducted on the anesthetised or sacrificed animal. SERS detection was then measured on the resected tissue to compare with histopathological validation [13]. Expansion of SERS detection capability has been achieved through coupling SERS with resonance Raman scattering and spatially offset Raman scattering (see below) to yield SERRS and SESORS, respectively. Further details on SERS-based techniques can be found in a review by Laing et al. [14].

### 3.3. Spatially Offset Raman Scattering (SORS)

SORS makes use of an applied spatial offset between the points of excitation and collection in a Raman measurement to collect photons that have been scattered by the subsurface medium. Comparisons between the spectra collected with no offset (surface measurements) and those collected with a lateral offset (subsurface measurements) enable the delineation of the spectral differences in composition at depth (Figure 4A) [15]. SESORS couples the sensitivity afforded by SERS with the subsurface probing of SORS to allow detection at even greater depths (~25 mm), with the potential for in vivo detection [7]. Recent examples include the detection of multicellular tumour spheroids in tissue [16] and glioblastoma multiforme (GBM) through intact skull in preclinical murine models (Figure 4B,C) [17].

### 3.4. Coherent Raman Scattering (CRS): Coherent Anti-Stokes Raman Scattering (CARS) and Stimulated Raman Scattering (SRS)

Coherent Raman techniques use two incident laser beams (termed the pump and Stokes lasers, respectively), where the frequency difference between the two beams is matched to a vibration of interest in the sample. The pump beam is usually a tuneable laser (typically 720–990 nm), whilst the Stokes beam is a fixed-wavelength laser (usually 1031 or 1064 nm). Retuning the pump laser to a different wavelength enables the detection of a different vibrational mode within the sample (Figure 5). Two prominent examples of CRS include coherent anti-Stokes Raman scattering (CARS) and stimulated Raman scattering (SRS). In CARS microscopy, the pump and Stokes lasers are tuned such that the frequency difference between them matches a molecular vibration of interest in the sample, which are then simultaneously directed onto the sample to generate a strong anti-Stokes Raman signal. Although the CARS signal is enhanced when the frequency difference between the pump and Stokes beams matches a chemical vibration, it also occurs under non-resonant conditions, resulting in a large background signal and a distortion of the CARS spectrum relative to the Raman spectrum of the target sample. These issues have been ameliorated by the development of stimulated Raman scattering (SRS) microscopy. SRS measures the stimulated emission of photons as either an intensity loss in the pump beam (stimulated Raman loss, SRL) or as an intensity gain in the Stokes beam (stimulated Raman gain, SRG) when the frequency difference between the pump and Stokes beams matches a vibrational mode within the sample. Therefore, SRS imaging typically acquires an image at a single vibrational frequency, for example, the CH_3_ vibration signifying the presence of proteins (those containing branches amino acids) within live cells (Figure 5). Retuning the pump laser enables the detection of alternative chemical species, for example, CH_2_ signifying lipid species (Figure 5). The resolution of SRS microscopy is currently around 400 nm in commercially available SRS microscope systems, and efforts to improve the spatial resolution are active areas of research [18]. SRS presents several advantages over CARS: the SRS spectrum replicates the spontaneous Raman spectrum, enabling quantification and straightforward qualitative analysis. Tagging strategies using alkynes and nitriles have proven fruitful for sensitive detection of a variety of chemical and biological species, including drugs [19,20], natural products [21,22] and sugars [23,24] in live cells and tissues.

## 4. Sample Preparation

Raman imaging techniques are generally compatible with direct imaging of live cells and tissue samples, requiring only minimal sample preparation. This is a major advantage because it offers a highly streamlined process for imaging. Equally, Raman imaging is compatible with fixed cellular samples, although caution must be exercised particularly when analysing formalin-fixed paraffin-embedded tissue samples. Formalin fixation-associated protein cross-linking can impact on protein bands within the region 1500–1700 cm^−1^ of the Raman spectrum (highlighted as the peak for amide-I (Figure 2)) [25]. In addition, paraffin wax is highly Raman-active, producing characteristic peaks generated by the C-C stretch (1134 cm^−1^) and CH_2_ deformation (1296 and 1444 cm^−1^), respectively [26]. Sample de-waxing strategies have been proposed to directly remove the paraffin wax from the sample, whilst digital processing to remove paraffin wax-related Raman spectra and features has also been proposed [26]. A set of representative SRS images of a tissue microarray are presented in Figure 5, along with a tissue core embedded in paraffin wax that shows a diffuse SRS signal at 2851 cm^−1^, that completely masks the underlying tissue signals, demonstrating the advantages of a de-waxing strategy.

In order to perform optimal SERS measurements, consideration must be made with respect to sample preparation and nanoparticle composition. Silver colloids are commonly used nanostructures. The Leopold and Lendl method is typically used to produce silver salts using a reducing agent such as hydroxylamine hydrochloride and sodium hydroxide (alkaline pH) [27]. Alternatively, silver nanoparticles can be synthesised using the sodium citrate reduction method [28]. Gold nanoparticles are typically prepared using a citrate reduction method reported by Turkevich et al. and can be prepared in a wide variety of shapes. A recent review by Langer et al. describes these in greater detail [29].

## 5. Spectral Analysis and Multivariate Techniques

The generated Raman spectrum includes peaks from all the Raman-active species within the sample, and therefore the technique can rapidly generate a large and information-rich dataset. When interfaced with a microscope system, the subsequent Raman images contain a full or partial Raman spectrum at each pixel within the image. As such, the extraction of chemical information may require multivariate analysis techniques to extricate the underlying chemical, biological or structural information. The analysis of subtle changes within the spectral profiles may be associated with disease, biological processes or the direct influence of external agents, for example, drugs. Salient techniques used in post-processing of Raman spectra include K-Means clustering analysis (KMCA), principal component analysis (PCA) and partial least squares regression (PLSR). In brief, KMCA aims to partition data into clusters based on the similarity of spectral features [30], whilst PCA is a method used to reduce the dimensionality of the dataset to describe the variation present within it [31]. PLSR aims to match a Raman spectral dataset to a series of validated targets, to investigate spectral variation as a function of systematic change (e.g., drug treatment [32]). Machine learning algorithms such as artificial neural networks have also received considerable attention for the analysis of bio-spectroscopy datasets, whilst recent advances have been reported using high-resolution SRS imaging datasets, including the use of spectral phasor analysis to delineate cellular compartments [33], and to investigate drug-induced effects [34]. A pertinent review conducted by Byrne et al. highlights post-processing analysis techniques applied to biological datasets in greater detail [35].

## 6. Raman Spectroscopy in Preclinical Prostate Cancer Models and Clinical Tumours

### 6.1. Detecting Prostate Cancer at the Tissue Level

Raman spectroscopy has been evaluated as a diagnostic tool to differentiate between malignant and non-malignant prostatic tissues. Aubertin et al. interrogated 32 fresh prostatectomy specimens using Raman spectroscopy to identify and grade prostate cancer [36]: 20–50 Raman spectra were acquired from each prostate slice, resulting in a total of 947 spectra. Raman spectroscopy identified prostate from non-prostatic tissue with a sensitivity of 82% and a specificity of 83%, which differentiated benign from malignant prostatic tissue with a sensitivity of 87% and a specificity of 86% (Figure 6) [36]. Of note, high Gleason grade (Grade 1–5) prostate cancer tends to have more pronounced Raman spectral profiles and can be identified more confidently, and the ability to detect cancer increased accordingly.

Intraoperative surgical guidance is an area of active research for clinical implementation of Raman spectroscopy. While restricted to small cohort studies, researchers have developed biopsy-needle-compatible probes to support real-time assessment of the tumour margins. Pinto et al. integrated Raman spectroscopy to the normal workflow of robotic-assisted radical prostatectomy for in vivo (*n* = 4) as well as ex vivo spectroscopic analysis (based on 599 spectra from 20 prostatectomy specimens, with an average of 30 spectra per patient, and grouped by anatomical regions) to differentiate between cancerous and non-cancerous prostate tissue [37]. Cancerous prostate tissue was distinguished from non-cancerous tissue with a sensitivity and specificity of 90.5% and 96%, respectively. The use of optical technologies such as Raman spectroscopy can provide an avenue for point-to-point characterisation in detecting cancer cells that have invaded beyond the anatomical confines of the prostate.

### 6.2. Treating Residual or Recurrent Microscopic Disease

Despite treatment with radical intent, approximately 20–40% of patients with clinically localised prostate cancer subsequently present with biochemical recurrence [38]. Microscopic tumours are not visible to the naked eye and may not be included as part of the prostatectomy specimen. Qiu et al. reported the use of a modified SERS approach as a platform for intraoperative eradication of microscopic tumours [39]. They developed a probe-based technology of gold particles embedded with Raman reporters that were readily taken up by prostate cancer cells following intravenous administration due to the enhanced permeability and retention effect [40,41]. In an orthotopic prostate cancer mouse model, upon the introduction of a high-power laser to the tumour, the administered nanostructures were able to localise to the tumour cells and generate a photothermal effect, resulting in micro-tumour ablation around the prostatic bed. Over a sixteen-day follow-up period, mice treated with surgical resection and Raman imaging-guided hyperthermia showed no local recurrence, whereas mice treated with standard surgical resection alone showed tumour growth in the first six days [39].

### 6.3. Raman-Based Analysis of Castration-Resistant Prostate Cancer (CRPC)

Over the last decade, novel inhibitors of the androgen receptor pathway have significantly improved the outcomes of patients with incurable prostate cancer [42]. Despite initial treatment response, adaptation by the tumour can result in relapsed disease. A better understanding of treatment resistance will help to identify novel therapeutic targets for castration-resistant prostate cancer (CRPC). A combination of proteomics, metabolomics and Raman spectroscopy were used to characterise a candidate biomarker, 2,4-dienoyl-CoA reductase (DECR1), involved in polyunsaturated fatty acid degradation and cell growth [43]. Raman spectroscopy confirmed enhanced tumoral lipid contents in CRPC [43]. This finding is consistent with an earlier study, whereby Raman spectroscopy was applied on a cohort of prostate cancer tissues (*n* = 50, 33 with hormone-naive disease and 17 with CRPC). Analysis of the Raman spectra was performed by machine learning algorithms and achieved a sensitivity of 88.2% and specificity of 87.9% in detecting CRPC [44].

### 6.4. Analysing Tumour Microenvironment of Lymphatics and Bony Metastasis

Pelvic lymph node dissection (PLND) is the procedure whereby pelvic lymph nodes at risk of harbouring metastatic deposits are surgically removed. Accurate detection of lymphatic metastasis is essential to guide treatment decisions. The standard of care nomograms to determine whether PLND will be performed suffer from both false negatives (underestimating the risk of nodal metastasis) and false positives (resulting in overtreatment) [45,46]. Occult metastatic pelvic lymph nodes can occur in 3–26% of patients who are negative on conventional cross-sectional imaging (e.g., magnetic resonance imaging or computerised tomography) [47,48,49]. PLND provides accurate staging information as well as therapeutic benefit [50]. Spaliviero et al. developed a method of detecting lymph node metastasis in a preclinical orthograft murine model using human PC-3 prostate cancer cells [11]. Infiltration of the medial iliac lymph nodes by PC-3 prostate cancer cells resulted in a decreased accumulation of injected surface-enhanced resonance Raman spectroscopy (SERRS) nanoparticles consisting of noble metals acting as an antenna to amplify the Raman scattering intensity. The metal particles were implanted into the dorsal lobes of the prostate, which later spread to the medial iliac lymph nodes and were internalised by macrophages. If a lymph node is either partially or completely invaded by prostate cancer cells, the macrophages are replaced by tumour tissue, resulting in a decrease in nanoparticle uptake [11].

Using a similar approach, Raman spectroscopy can be used to detect bony metastasis in vivo. Bone metastasis occurs in 8–35% of newly diagnosed prostate cancers [51,52,53]. Radionuclide bone scan is limited by low specificity, high cost and radiation damage [54]. More recently, the performance of PSMA PET/CT in detecting distant metastasis has resulted in its adoption as a standard of care imaging modality in patients with suspected metastatic disease. The application of PSMA PET imaging for the detection of recurrent prostate cancer and metastatic CRPC remains the subject of ongoing research [55]. Novel applications of SERS may further improve the sensitivity and specificity in detecting recurrent and/or metastatic prostate cancer. Shao et al. assayed collagen in human serum by SERS. A total of 427 patients with prostate cancer were included in the study (204 with and 223 without skeletal metastases) [56]. A type of machine learning algorithm known as convolutional neural network was developed to extract features of the Raman spectra associated with the presence of bone metastasis. The Raman ‘signature’ identified patients with bone metastasis with a sensitivity of 80% and specificity of 83% [56].

Understanding the mineralisation of bone is equally important in detecting metastasis. In a severe combined immunodeficient mouse model, bone metastasis was introduced by direct injection of human prostate cancer (LNCaP or its derived C4–2b subclone) cells into the tibia of the mice. Raman spectroscopy-based analysis revealed significant changes to bone mineralisation in the cortex of tumour-bearing tibia. The ratio of carbonate:matrix of bone is an important parameter associated with tissue maturity and disease. In bones affected by prostate cancer, the carbonate:matrix ratio significantly decreases, suggesting elevated bone turnover (*p* < 0.001) [57]. The compositional alterations were accompanied by architecture deterioration observed on CT scan and radiographical imaging.

### 6.5. Raman-Based Analysis to Characterise Altered Lipid Metabolism in Prostate Carcinogenesis

Recent advances in metabolism show that prostate cancer cells are capable of reprogramming their mitochondria to increase the synthesis of lipids [58]. Raman spectroscopy has been used to elucidate the status of lipid metabolism. Roman et al. tested the feasibility of time-dependent Raman spectroscopy to investigate lipid droplet composition in untreated and X-ray-irradiated human prostate cancer PC-3 cells. Besides the variation of lipid concentration between cells, intracellular lipid droplets exhibited significant heterogeneity in composition (containing varying amounts of triacylglycerols and cholesteryl esters). Accumulation of cytoplasmic lipid droplets occurred 48 h after high-dose X-ray radiation, consistent with apoptosis-induced lipid accumulation [59]. Of note, upregulation of key lipogenic genes was previously shown to be associated with prostate cancer progression and decreased overall survival in patients [60].

Jamieson et al. investigated the effects of propranolol, a beta-blocker, on lipid metabolism in prostate cancer cells. Treatment with propranolol resulted in lipid accumulation in PC-3 cells compared to non-cancerous PNT2 cells [5]. The adrenergic system is activated in times of stress or starvation, mobilising and degrading lipids for ketone body formation. Adrenergic-stimulated peripheral lipolysis requires autophagy. Autophagy is the catabolic process essential for cell survival under times of stress. Propranolol can directly inhibit autophagy in prostate cancer cells [61,62], which explains why lipid accumulation was observed as a by-product.

### 6.6. Analysis of Biofluids as Liquid Biopsies in Detecting Prostate Cancer

In the earlier sections of this review, we have focused on the application of Raman spectroscopy techniques on disease diagnosis in tissues. However, clinically relevant diagnostics have also been investigated for samples removed from the body, in particular pertaining to blood analytes. Blood analytes include cell-free DNA (cfDNA), circulating tumour cells (CTCs) and extracellular vesicles (EVs).

Medipally et al. compared mean Raman spectra of plasma from 43 patients with prostate cancer and 33 healthy volunteers. An increase in the bands associated with nucleic acids from cell-free DNA was observed in the Raman spectra of plasma from prostate cancer patients [63], whereby increased nucleic acids may arise from either upregulated gene expression or increased cellular release due to cell death, such as apoptosis. In addition, the bands corresponding to lipids were more abundant in the plasma of prostate cancer patients compared to healthy individuals. Of note, increased tumoral lipid and cholesterol abundance are associated with aggressive and treatment-resistant disease [64,65].

Raman spectroscopy has also been employed to generate spectral fingerprints from extracellular vesicles (EVs) released by prostate cancer cells. EVs are small, lipid-bound particles containing nucleic acid and protein, which are released by metabolically active cells into the tumour microenvironment, with some EVs found in the circulation. EVs function to support communications between tumour cells and their local environments [66]. Lee et al. compared Raman signatures from red blood cell and platelet-derived EVs of healthy patients and the prostate cancer cell lines (namely PC-3 and LNCaP cells). EVs from malignant cells tended to have increased lipid and protein contents compared to control EVs [67].

Raman spectroscopy is highly versatile and can be used to analyse urine samples in addition to blood. Del Mistro et al. carried out a proof-of-concept SERS analysis on urine samples from 18 patients (9 with prostate cancer scheduled to undergo radical prostatectomy and 9 healthy controls). Using a machine learning algorithm, the spectral data were classified into two classes (prostate cancer or healthy individuals). Only one sample was misclassified by this model, leading to a sensitivity of 100% and a specificity of 89% [68]. Equally, Ma et al. deployed SERS to analyse the mean spectra of urine samples from 75 patients (12 patients with recurrent prostate cancer and 63 patients with recurrence-free prostate cancer). The data showed enhancement in Raman bands associated with lipids, proteins, amino acids and DNA in patients with recurrent prostate cancer compared to recurrence-free patients [69]. Further details on urine-based Raman techniques can be found in a review by Chen et al. [70].

One of the most common clinical tests to identify men at risk of prostate cancer is measurement of the serum level of prostate-specific antigen (PSA) protein. Unfortunately, the PSA test has its limitations, commonly associated with type I errors [71]. It can be falsely elevated in non-malignant conditions, such as benign prostatic hyperplasia (BPH), prostatitis and urinary tract infections [72]. Chen et al. utilised SERS of blood serum as a screening tool to differentiate between prostate cancer and benign prostatic hyperplasia (BPH) in males with prostate-specific antigen levels in the 4–10 ng/mL range, which represents a major diagnostic challenge [73]. A total of 240 spectra were acquired from 40 prostate cancer patients and 40 BPH patients with diagnosis based on prostatic biopsies. Spectra-based analysis offered a sensitivity of 97.5% and a specificity of 100% in differentiating prostate cancer from BPH [73]. Using a high-throughput method, Medipally et al. designed a novel methodology for using Raman spectroscopy on blood samples from 10 prostate cancer patients and 10 healthy volunteers and reported a sensitivity and specificity of 96.5% and 95%, respectively, in identifying prostate cancer patients [74].

## 7. Discussion

Raman imaging techniques are optical methods which are minimally invasive and typically non-destructive when conducted under optimised conditions. Given that Raman imaging techniques provide chemically specific detection, often without the requirement for contrast agents, there remains a huge potential for in vivo applications across the remit of disease diagnosis, monitoring therapeutic outcomes and real-time surgical guidance. With appropriate design, SERS nanoparticles have shown efficient sensitivity and specificity for disease diagnosis in a variety of biofluids.

Technological advances in lasers and optical components have been realised over the past decade, and to that end, relatively inexpensive, off-the-shelf infrared Raman devices are already used to differentiate between malignant and benign tissue in breast cancer [75]. Furthermore, innovations in medical appliances are creating opportunities for the translation of Raman spectroscopy for in vivo analysis. For example, long-lasting, durable and sterilisable fibre optic probes are now available and may be used for intraoperative analysis. Lastly, digital analysis and chemometric techniques are continually being refined for the interpretation of Raman spectra and images with relative ease. Together, these developments will help to define how best to apply Raman spectroscopy-based techniques in addressing unmet clinical needs. Raman spectroscopy can enable intraoperative diagnosis and selective dissection of tissue. A few systems based on Raman spectroscopy have already been successfully commercialised; for example, the Progeny spectrometer (Rigaku Rama Technologies Inc., Tokyo Japan) has been used to detect colorectal cancer [76].

Considerable potential exists for the use of fast-acquisition SRS imaging for surgical applications. As a label-free optical imaging method, based on differences in the Raman spectral profiles, SRS has been successfully applied to differentiate tumours against non-neoplastic tissue in a glioblastoma murine model [77], along with seminal data that demonstrated the correlation between SRS imaging and conventional histology in the detection of glioma infiltration. Furthermore, multimodal imaging approaches will offer additional advantages for clinical translation. As a pertinent example, SRS and fluorescence imaging are often combined to offer multiplex detection of different regions within cellular and clinical samples [78]. The complementarity of these techniques facilitates the detection of a greater diversity of features within the sample, thus enhancing the information available to the scientists and clinicians. For instance, combining Raman imaging with fluorescence and brightfield microscopy may help to facilitate the introduction of these new modalities into existing clinical and histological workflows [79].

For diagnostic applications, SERS-based techniques have shown considerable utility within prostate cancer disease. However, point-of-care SERS devices are currently limited by the requirement for trained personnel to conduct the analysis, the need for specialised equipment and limits of detection that fall below the required safety standards for applications in medical settings [80]. However, significant effort is directed within industrial and academic settings to refine the hardware and sample workflow of SERS devices for disease detection, with some early success reported towards multiplex detection of Ebola, Lassa and Malaria [81]. The development of microfluidic devices to facilitate SER(R)S detection is an active area of development and may also facilitate the translation of the technique into clinical workflows. The diagnosis of prostate cancer using SORS measurements at depths for potential cancerous sites within the prostate poses a potential research area of significant interest. Current detection depths are within the millimetre range, and therefore, beyond the current scope of in vivo prostate cancer detection. However, interfacing SERS probes with SORS detection could overcome this limitation. Where a SERS reporter with a strong Raman signal is used, detection at greater depths can be achieved.

Despite its advantages and potential, Raman spectroscopy is yet to be widely implemented in clinical practice (Table 3). This may be due to its long spectral acquisition time required to improve the signal-to-noise ratio. The acquisition time for a Raman spectrum depends on several parameters, including the scattering tendency of a sample, the volume of a sample, the spectral quality required and the wavelength selected. A shorter wavelength generates a greater Raman scattering effect at the expense of potential phototoxic effects. However, a single spectrum can take up to a few seconds to acquire. Although short in duration, it is often necessary to amalgamate a large number of spectra over an area of a cell or tissue to gain a greater understanding of the underlying biology. Creating an image can take several minutes and up to a few hours, which would be too slow for some live cell imaging applications. The advent of slit scanning Raman imaging has improved acquisition times [82], whilst SRS imaging has shown that video-rate imaging speeds are achievable in biological samples [83].

Considering the technological advances that have been realised in Raman microscopy systems over the past ten years, there remain challenges regarding microscope design and subsequent regulatory approval before systems can be deployed in the clinic. For example, many coherent Raman imaging systems are either a bespoke creation built by research groups or are formed by combining relevant, commercially available components. As such, these systems are not conducive for clinical application. Commercial systems which offer the complete multiphoton imaging capability with built-in coherent Raman imaging have recently been launched to the market [84] and may begin to address this limitation. However, there remains significant technological developments to be addressed before these systems can be used in real-world clinical applications. Clinical Raman imaging using SERS nanoparticles is not currently widely applied. Some important underlying reasons are that, unlike other optical imaging modalities such as fluorescence, SERRS nanoparticles are not commercially available and instrumentation for Raman imaging is currently not widespread [13]. However, current research is aiming to address these limitations to facilitate the translation of Raman-based techniques into clinical settings.

## 8. Conclusions

Within the capability of non-destructive, non-invasive and highly sensitive analysis of biomolecules, Raman spectroscopy-based techniques offer a reliable tool for cancer detection. The spectral information acquired with Raman spectroscopy is beyond the reach of other techniques, especially if applied under in vivo conditions. Testing the potential of Raman spectroscopy for clinical benefit will be a major scope for the next 10 years. In the future, it is foreseeable that there will be a major push for intraoperative use and/or multimodal imaging approaches that utilise the strength of each spectroscopic modality to improve prostate cancer detection and treatment.

## Figures and Tables

**Figure 1 cancers-14-01535-f001:**
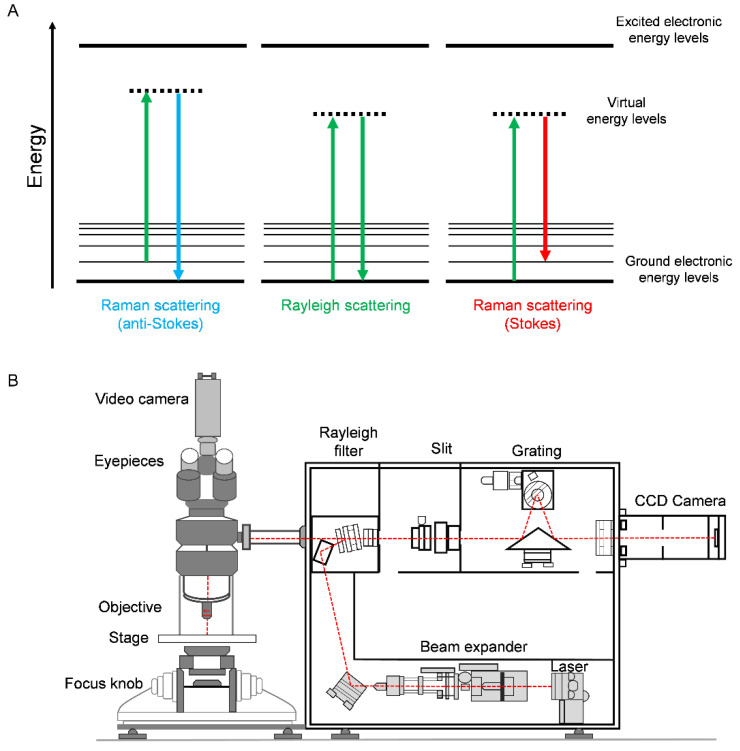
Applications of Raman spectroscopy in prostate cancer disease. (**A**) Energy level diagrams for light scattering. In Rayleigh scattering, photons are scattered with the same energy, and hence frequency, as the incident photons. Raman scattering results in an energy transfer from the incident photons to the chemical bond (Stokes Raman scattering) or from the vibrationally excited chemical bonds to the incident photons (anti-Stokes Raman scattering). Thus, Stokes Raman scattering results in photons with a lower energy than the incident photons, whilst anti-Stokes Raman scattering results in photons of a higher energy than the incident photons. (**B**) A schematic diagram of a Raman microscope. Raman microscopes consist of the following components: a laser source generating monochromatic light, a microscope setup with an objective lens to focus the laser beam and sample stage to enable mapping of the sample (the objective lens also collects the Raman scattered photons in backscattering configuration), a Rayleigh filter to remove the Rayleigh scattered light (the excitation wavelength), a pinhole for confocal sectioning of the target sample which removes out-of-focus light, a diffraction grating which separates the photons according to wavelength, and typically, a charge couple device (CCD) detector to capture the intensity of photons at each wavelength.

**Figure 2 cancers-14-01535-f002:**
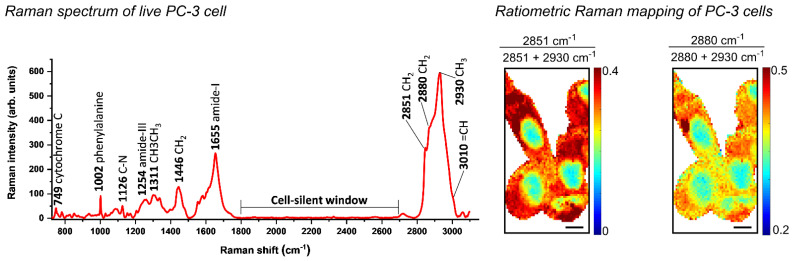
Raman spectroscopy of PC-3 cells. The Raman spectrum was acquired from a live PC-3 cell using a 532 nm laser excitation source, which was focused onto the sample using a 60× objective lens (18 mW) for 10 s. Ratiometric Raman mapping of live PC-3 cells using the same acquisition settings except spectra were acquired for 0.5 s per pixel, with a 1 μm pixel size for imaging. The images present the lipid/protein ratios using the following Raman bands: 2851 cm^−1^ (CH_2_ symmetric stretch, lipids), 2880 cm^−1^ (CH_2_ asymmetric stretch, lipids and proteins) and 2930 cm^−1^ (CH_3_ symmetric stretch, proteins), based on formulae accompanying each panel. Nuclear regions are identified with a lower ratio than the surrounding cytoplasm, which is lipid-rich. Scale bars: 10 μm.

**Figure 3 cancers-14-01535-f003:**
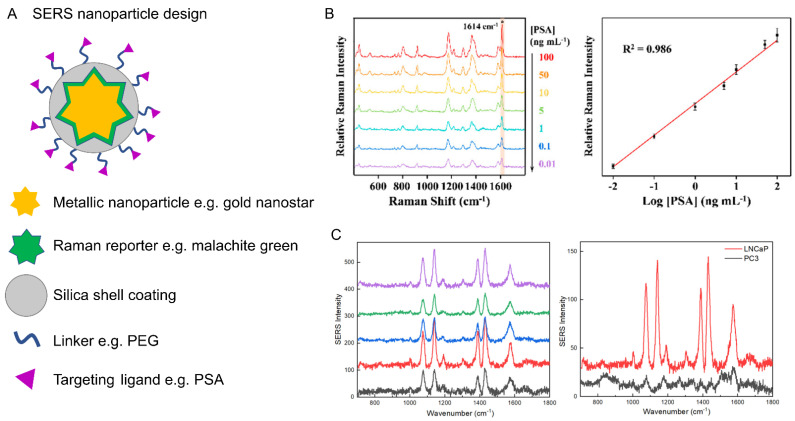
Biomolecular targeting using SERS detection. (**A**) Conceptual design of SERS nanoparticles for biological analysis. Typically, a gold nanoparticle, in this case a nanostar, is coated in a Raman reporter molecule. The reporter usually has a large Raman cross-section and may have an absorption matched to the excitation wavelength for enhanced detection sensitivity via resonance coupling effects. The nanoparticle may be coated in a thin silica shell, which stabilises the gold core and also provides a matrix for the Raman reporter. Specific targeting is achieved through a targeting ligand, e.g., antibodies, DNA, micro-RNA, etc., which can be introduced through a linker group. (**B**) Detection of prostate-specific antigen (PSA) using a microfluidic device with a SERS limit-of-detection at 0.01 ng/mL. Adapted from [9] with permission from the American Chemical Society (copyright 2019). (**C**) Detection of prostate-specific membrane antigen (PSMA) using SERS nanostars functionalised with 4-aminothiophenol in five different LNCaP cancer cells, which express PSMA, whilst a specificity comparison with SERS spectra from individual LNCaP and PC3 cells (which do not express PSMA) is also provided. Adapted from [10] with permission from the American Chemical SocietySociety (copyright 2018).

**Figure 4 cancers-14-01535-f004:**
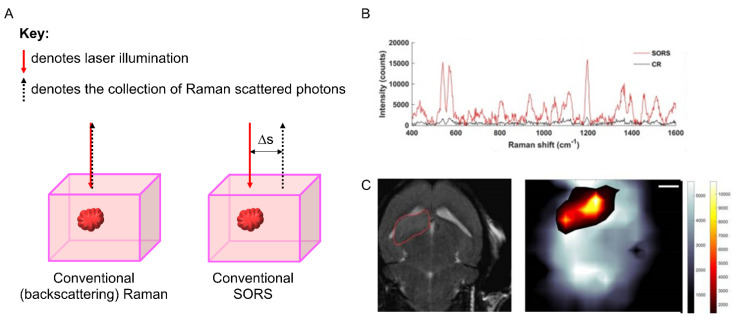
(**A**) Schematic diagram detailing the excitation and collection configuration for conventional (backscattered) Raman, where the laser excitation and Raman scattered photon collection are at the same point. In a conventional SORS experiment, the point of collection is offset from the excitation by a spatial offset (Δs). Adapted from [15] with permission from the Royal Society of Chemistry (copyright 2021). (**B**) Non-invasive in vivo imaging of integrin-targeting SERRS nanoparticles through the skull in GBM-bearing mice by means of conventional Raman (CR) and SORS. Representative CR and SORS spectra from a point of maximum intensity of the SERRS NPs are provided in (**B**). (**C**) 2D axial T2-weighted MRI scan confirming the presence of a left frontal tumour (outlined in red). SORS heatmap of the bone (greyscale) and SORS heatmap of the SERRS NPs (red hot) are overlaid and correlate with the tumour region highlighted in the MRI scan. Images reproduced from [17] with permission from Ivyspring International Publisher (copyright 2019).

**Figure 5 cancers-14-01535-f005:**
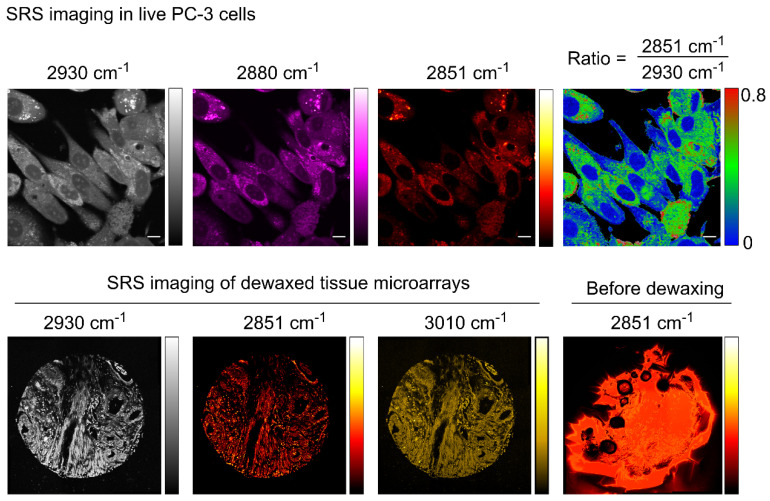
SRS imaging of live PC-3 cells. Images were acquired at 2930 cm^−1^ (CH_3_ symmetric stretch, proteins), 2880 cm^−1^ (CH_2_ asymmetric stretch, proteins and lipids) and 2851 cm^−1^ (CH_2_ symmetric stretch, lipids). A ratiometric image of the CH_2_/CH_3_ (2851 cm^−1^/2930 cm^−1^, intensity scale 0–0.8 a.u.) is provided which resolves the nucleus (<0.2) and lipid droplets (>0.7) within the cells. Scale bars: 10 μm. SRS imaging of a de-waxed tissue microarray (TMA) is also presented. SRS images were acquired at 2930 cm^−1^ (CH_3_), 2851 cm^−1^ (CH_2_) and 3010 cm^−1^ (=CH). A wax-embedded TMA core imaged at 2851 cm^−1^ (CH_2_) shows signal intensity across the core arising from the wax.

**Figure 6 cancers-14-01535-f006:**
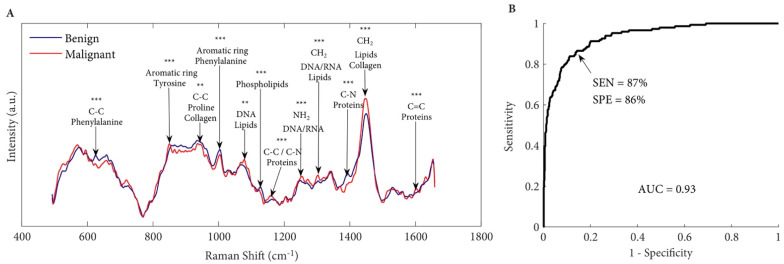
(**A**) Average Raman spectra for benign (*n* = 776) and malignant (*n* = 149) tissues. Corresponding peaks are shown with black arrows. (**B**) Receiver operating characteristic (ROC) curve distinguishing benign from malignant tissue with an accuracy of 86%. SEN, sensitivity; SPE, specificity; AUC, area under the curve. Univariate analysis performed on Raman peaks are illustrated with *p* values: *** *p* < 0.001, ** *p* < 0.01. Images reproduced from [36] with permission from Wiley (copyright 2018).

**Table 1 cancers-14-01535-t001:** Glossary of Raman Spectroscopy modalities.

Terms	Definitions
Rayleigh scattering	Rayleigh scattering is an elastic scattering process where there is no change in the energy of the photons upon interaction with the target sample.
Stokes/anti-Stokes Raman scattering	Raman scattering (or inelastic scattering of incident photons) may result in either anti-Stokes scattering when there is an energy gain or Stokes scattering when there is an intensity loss with respect to the incident photons following interaction with the molecular vibration.
Resonance Raman spectroscopy	The frequency of the excitation laser source matches (or approaches) that of an electronic transition within the target sample, thus enhancing the Raman scattering signal.
SERS—Surface-enhanced Raman scattering	A technique for molecular detection that relies on the enhanced Raman scattering of molecules that are adsorbed on, or in close proximity to, SERS-active metal surfaces, including gold or silver nanoparticles.
SORS—Spatially offset Raman scattering	Low-intensity laser excitation is directed onto the surface of the sample, and Raman spectra are obtained at a known spatial offset from the illumination spot. By applying a spatial offset to the detection, it enables the collection of Raman scattered photons that have been produced at greater (and variable) depths within the sample material.
Coherent Raman scattering	Raman-active vibrations can be selectively driven into coherence by exciting with two (or more) laser wavelengths. The frequency difference of the excitation sources is matched to a vibrational resonance of the target molecule, thus enhancing its detection. Coherent anti-Stokes Raman scattering (CARS) and stimulated Raman scattering (SRS) are examples of coherent Raman scattering.

**Table 2 cancers-14-01535-t002:** Advantages and disadvantages of Raman spectroscopy for biomedical diagnostics.

Advantages	Disadvantages
Non-destructive, non-invasive High specificity Simultaneous detection of biomolecules Compatible with physiological measurements due to minimal water interaction In vivo applications Suitable for chemical analysis, quantification, classification and imaging of biological materials	Weak Raman signals can lead to long acquisition times Not widely incorporated into current clinical workflows Sophisticated data analysis Autofluorescence can overwhelm the Raman signal (sample dependent)

**Table 3 cancers-14-01535-t003:** Advantages and disadvantages of Raman spectroscopy and its subtypes.

Technique	Advantages	Disadvantages
Spontaneous Raman scattering	Minimal sample preparation and setup Minimal interference with water for biological investigation in live cells	Autofluorescence from cells and tissues can overwhelm Raman signals Image acquisition can be slow (seconds to minutes) for some applications
Resonance Raman scattering (RRS)	Improved sensitivity and selectivity over standard Raman spectroscopy	High-fluorescence background signal capable of obscuring true Raman signals
Surface-enhanced Raman scattering (SERS)	Improved sensitivity and selectivity over standard Raman spectroscopy	Requires coupling with nanoparticles SERS nanoparticles must be biocompatible if used in vivo
Spatially offset Raman scattering (SORS)	Delineation of spectral differences in composition at greater depths In vivo detection through tissue	Complex setup and hardware required
Surface-enhanced spatially offset Raman spectroscopy (SESORS)	Couples the sensitivity afforded by SERS with subsurface probing of SORS Detection at greater depths can be achieved	Requires active targeting of SERS nanoparticles for detection
Coherent anti-Stokes Raman scattering (CARS)	Fast image acquisition Typically uses biocompatible NIR laser excitation Minimal background fluorescence	Requires tuneable lasers to probe molecular structures, which are expensive Non-resonance signal distorts the CARS spectrum
Stimulated Raman scattering (SRS)	Enhanced signal strength compared to spontaneous Raman scattering Fast image acquisition (μs/pixel) Biocompatible NIR excitation SRS spectrum matches Raman spectrum for easy peak assignment and quantification	Requires tuneable lasers to probe molecular structures, which are expensive Some background signal can be detected from pump-probe-based effects Complex hardware makes it difficult to incorporate into a handheld device for intraoperative use

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
