# Peer review of "Raman Spectroscopy in Prostate Cancer: Techniques, Applications and Advancements"

_cancers, 2022, doi:10.3390/cancers14061535_

Round 1
Reviewer 1 Report
This review summarizes different Raman spectroscopic techniques and their applications in prostate cancer. Potentially, the review topic is of general interest to the readers of the journal; however, it requires substantial changes and reevaluation of the literature to be suitable for publication. A recent review published in Cancers (Xu, J. et al. Unveiling Cancer Metabolism through Spontaneous and Coherent Raman Spectroscopy and Stable Isotope Probing. Cancers, 2021) covers a general and broad introduction of applications of Raman techniques to cancer. The main objective of this review should more specifically focus on the subject of prostate cancers; however, the number of literature cited in this review is very minimal, and the selection of them seems random. A comprehensive review of the same topic has been published (Kast, R. E. et al. Emerging technology: applications of Raman spectroscopy for prostate cancer. Cancer Metastasis Rev, 2014), and it should be critically read and cited. To substantially modify the manuscript, a more systematic evaluation of the literature should be done, and differences of this review compared with other reviews should be highlighted.
Other points:
- A comparison between Raman spectroscopy and other techniques used to study prostate cancer would be useful, instead of/in addition to in the current manuscript a comparison made between Raman techniques.
- The number of figures should be increased. Some exemplified figures in cited literature can be included.
Reviewer 2 Report
The work is devoted to a review on a very important topic, the study of prostate cancer using various methods of Raman spectroscopy and their application in clinical practice.
Major comments:
- The object of research is prostate cancer, which is common and dangerous, this is understandable. And what about its classification, when should it be detected, at what stage? What is a marker for prostate cancer? What is the main medical task in the study of prostate cancer and what role can Raman spectroscopy play in this task?
- For a review, and even on such an extensive topic, in my opinion, there are not enough links. For example, in a 2014 review on a similar topic, Cancer Metastasis Rev. 2014 Sep;33(2-3):673-93. doi: 10.1007/s10555-013-9489-6 222 refs were used.
- Few illustrations. There is no illustration of schemes of Raman spectroscopy techniques, difficult to understand, especially for non-specialists in the field of Raman spectroscopy. It may be useful to present the information in the form of a table.
- Only one Raman spectrum of prostate cancer cells. Why are there no other spectra, how different are the spectra of cancer from the control group. What are the spontaneous spectra of tissues and biofluids, SERS, SORS, SRS spectra?
- What are the limitations of Raman spectroscopy (long analysis time, complexity of measurements, high cost, high qualification of the operator)? Why this method is not yet in clinical practice for the diagnosis of prostate cancer.
- How can SERS be used for in vivo tissue studies in combination with SORS? How will the injection of metal nanoparticles into the body be dealt with, how will they be excreted from the body? Could the nanoparticles used for Raman spectroscopy be toxic? What local effect can they have on tissues?
- There is no interpretation of the bands of the Raman spectra and their relationship with biochemistry. Now only substances are in the review, but what do they represent in the spectrum? What are these bands, are they always the same or different, one Raman band or several, absolute or relative intensities, etc.?
Minor comments:
- lines 59-60 "As such, a Raman spectrum is a plot of the photon intensity in arbitrary units (a.u.)" why arb. units?
- figure 1, on figure notch filter, but in text only filter, what is the difference? Maybe you should exclude the "notch" grom figure. Where is a laser in the figure? fig 1.D "intensity scale 0-0.8 a.u." where is this scale in the figure?
- line 89 "which reduces the background interference in the resultant spectra", "background interference" - how is it possible?
- line 107 - Why is there so little information about spontaneous Raman spectroscopy? From this paragraph, one gets the impression that it has only flaws and is not used anywhere in medical research. Although this is not the case, especially since in this review there are many examples of the successful use of spontaneous Raman spectroscopy for the study of the prostate.
- lines 223-224 - no separation accuracy (sensitivity/specificity) values.
Reviewer 3 Report
This review “Raman spectroscopy in prostate cancer: techniques, applications and advancements” by Fortis Gaba et al. provides a focus on the preclinical and clinical achievements in implementing Raman-based techniques, highlighting remaining challenges for clinical applications.
The paper is within the scope of Cancers, while some details are unclear. I hope the following comments may help to improve the quality of the paper for future submission.
1. It is suggested to supplement the application of nanoplasma materials in surface-enhanced Raman scattering enhancement.
2. In P5, Sample preparation, it is better to add some other types of sample processing methods. E.g., for biofluids, some pretreatment methods such as dilution, acidification/alkalization, centrifugation, filtration, or deproteinization, were most commonly used before SERS analysis.
3. There has also been great progress in the development of microfluidics for use in in vitro diagnostics. It is suggested that the recent advances in the integration of a SERS modality with different types of microfluidic devices should also be included in this review.
4. In P5, Spectral analysis and multivariate techniques. It is recommended to describe the three methods (PCA、KMCA、PLSR) in more detail, including advantages/disadvantages, differences among them, as well as specific application conditions. In addition, it is suggested to cite the references of these three multivariate data analysis techniques
5. On page 2, the units in line 50 (Raman Shift = 0 cm-1) was incorrectly written. cm-1 ---> cm-1. Similarly, the units in line 94, 95, and 97 on page 4 also need to be corrected.
Round 2
Reviewer 1 Report
The authors have nicely addressed all my concerns from last time and made substantial revisions accordingly. The authors have also justified their logic for literature selection. This manuscript is of general interest to the readers of the journal and with good quality.
Reviewer 2 Report
The authors of the work responded in detail to all comments on the work. My opinion - the work can be published in this form. Only the manuscript with red corrections was attached, the text should be checked by the editor.
Reviewer 3 Report
All questions were answered and the manuscript was modified following the reviewer's comments, thus I recommend publication of the manuscript.